

# Spatiotemporal Bedload Transport Patterns Over Two-Dimensional Bedforms

Kate C.P. Leary [1], Leah Tevis [1], and Mark Schmeeckle [2]

[1]Department of Earth Environmental Sciences, New Mexico Tech, Socorro, NM
[2]School of Geographical Sciences and Urban Planning, Arizona State University, Tempe, AZ

**Correspondence:** Kate Leary (kate.leary@nmt.edu)

**Abstract.** Despite a rich history of studies investigating transport over bedforms and dunes in rivers, the spatiotemporal patterns of sub-bedform bedload transport remain poorly understood. Previous experiments assessing the effects of flow separation on downstream fluid turbulent structures and bedload transport suggest that localized, intermittent, high-magnitude transport events (i.e., permeable splat events) play an important role in both downstream and cross-stream bedload transport near flow

reattachment. Here, we report results from flume experiments that assess the combined effects of flow separation/reattachment and flow reacceleration over fixed, two-dimensional bedforms (1.7 cm high; 30 cm long). A high-speed camera observed bedload transport along the entirety of the bedform at 250 f/sec. Grain trajectories, grain velocities, and grain transport direction were acquired from bedload images using semi-automated particle tracking techniques. Downstream and vertical fluid velocity was measured 3 mm above the bed using Laser Doppler Velocimetry (LDV) at 15 distances along the bedform profile.

Mean downstream fluid velocity increases nonlinearly with increasing distance along the bedform. However, observed bedload transport increases linearly with increasing distance along the bedform, except at the crest of the bedform, where both mean downstream fluid velocity and bedload transport decrease substantially. Bedload transport time series and manual particle tracking data show a zone of high-magnitude, cross-stream transport near flow reattachment, suggesting that permeable splat events play an essential role in the region downstream of flow-reattachment.

# 1   Introduction

Although bedload transport has been a subject of scientific inquiry for over a century (Gilbert, 1877; Gilbert and Murphy, 1914), our understanding of bedload transport mechanics on a sub-bedform scale remains limited (Terwisscha van Scheltinga et al., 2021; Leary and Schmeeckle, 2017). Sub-bedform transport mechanics potentially play an essential role in the calculations of bedload transport and our understanding of the three-dimensionality of bedform evolution. Sub-bedform particles are not

stationary but have detectable mean velocities (Ashley et al., 2020). However, relatively few studies have focused on sediment transport patterns on a sub-bedform scale due to the difficulty in measuring particle migration (Terwisscha van Scheltinga et al., 2021; Radice, 2021). Due to the dearth of studies on this subject, it is critical to start at first principles and assess bedload transport dynamics associated with the two primary and fundamental fluid regimes of bedforms: flow separation/reattachment and flow reacceleration. An abundance of experiments of this nature has been conducted in the latter half of the 20th century



(Vanoni and Nomicos, 1960; Raudkivi, 1963, 1966; Vanoni and Hwang, 1967; Rifai and Smith, 1971; Vittal et al., 1977; Itakura and Kishi, 1980; Van Mierlo and De Ruiter, 1988; Nelson and Smith, 1989; Wiberg and Nelson, 1992; Lyn, 1993; Nelson et al., 1993; McLean et al., 1994; Nelson et al., 1995; Bennett and Best, 1995). However these studies primarily focused on fluid mechanics over bedforms and did not always analyze bedload transport at high spatiotemporal resolutions. Previous work is therefore not accounting for the amount of sediment being transported in the streamwise and cross-stream
directions on a sub-bedform scale (Leary and Schmeeckle, 2017; Unsworth et al., 2018).

Previous studies have focused on the sub-bedform spatiotemporal patterns of turbulent fluid structures but were limited in their sediment transport analyses. Rather than looking at the spatiotemporal patterns of sediment transport, these studies were limited to whether specific turbulent structures induced entrainment or not. Bennett and Best 1995 found that the turbulence structure over bedforms is intrinsically linked to the flow separation zone's development, magnitude, and extent, which can have

a wide variation among neighboring bedforms (Terwisscha van Scheltinga et al., 2021). Notably, they observed the importance of quadrant 4 events near flow reattachment as significant contributors to the local Reynolds stress and sediment entrainment. In agreement with McLean et al. 1994, Bennett and Best 1995 also indicate that, in addition to quadrant 4 events, quadrant 1 events may play an important role in entrainment near flow reattachment. Investigations such as those conducted by Bennett and Best 1995 provide important sub-bedform scale observations regarding fluid turbulence and bedload transport over bedforms.

However, the advent of new technologies and methods, particularly semi-automated particle tracking techniques and higher precision numerical models, suggest the need for experimental replication. Experiments by Tsubaki et al. 2018 showed that under turbulent flow conditions, as seen in the trough area of a bedform, particles experience a specific behavior of transport not seen along other areas of the bedform. Various experimental work shows that particle behavior depends on the location and position relative to the bedform in question (Ashley et al., 2020).

The flume and numerical experiments of fluid and bedload dynamics downstream of a backward-facing step by Leary and Schmeeckle 2017 and Schmeeckle 2015 assessed the effect of flow separation and reattachment to downstream bedload and fluid dynamics. Terwisscha van Scheltinga et al. 2021 noted in their bedform experimentation that the transport occurring on the lower stoss side of the dune was comparable to the transport seen in the backward-facing step experiments. Once the flow begins to accelerate along the stoss side of the bedform, this is where the difference in transport over a backward-facing step

and river dune can be seen.

Schmeeckle (2015) and Leary and Schmeeckle (2017) found distinct fluid turbulent structures near flow reattachment called splat events. Splat events are localized, high magnitude, intermittent flow features in which fluid impinges on the bed, infiltrates the top portion of the bed, and then exfiltrates in all directions surrounding the point of impingement, initiating bedload transport in a radial pattern (Perot and Moin 1995; Figure 1). These turbulent structures are primarily associated with quadrant

1 and 4 events (Schmeeckle, 2015; Leary and Schmeeckle, 2017). Splat events generate a distinct pattern of bedload transport compared to transport dynamics distal to flow reattachment. Unlike splat events, distal-to-flow reattachment bedload transport is characterized by unidirectional transport Leary and Schmeeckle (2017).

The investigations described above indicate that splat events play an essential role in the initiation and pattern of bedload transport proximal to flow reattachment. It is unclear, however, if these events remain an important factor in bedload transport





# Splat Event

**A** **Side View: Fluid Dynamics of Splat Event**

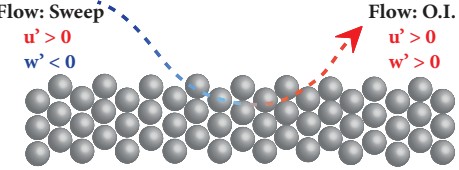

**B** **Map View: Bedload Dynamics of Splat Event**

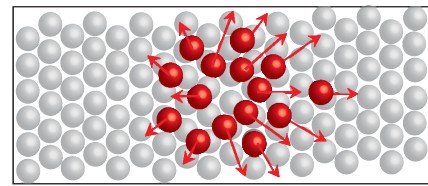

**Figure 1.** Schematic of fluid and bedload dynamics associated with a permeable splat event from Leary and Schmeeckle (2017). (A) Flow characterized by positive streamwise velocity fluctuations and negative vertical velocity fluctuations (sweep turbulent structure) impinges on and penetrates into the bed. This causes exfiltration in all directions around the point of infiltration, characterize by flow with both positive vertical and streamwise velocity fluctuations (outward interaction, O.I., turbulent structure). This initiates bedload transport by ejecting grains from the bed in all directions (B).

when full bedforms are present. Furthermore, do splat events continue to play a role in bedload transport when both flow reattachment and flow reacceleration are present? What is the overall pattern of sediment transport over a two dimensional bedform? To assess these questions, a series of flume experiments were run in which bedload motion and fluid velocities were observed over stationary ripples. By understanding the evolution of bedforms and grain transport in fluvial systems better, we can accurately develop predictive river models and plan for future water resource use (Guala et al., 2020).

# 2 METHODS

## 2.1 EXPERIMENTAL METHODS

Experiments were conducted in the sediment transport research flume at the US Geological Survey's Geomorphology and Sediment Transport Laboratory in Golden, CO. This recirculating flume is approximately 6m x 0.25m. The flume was lined with 17 cement ripples, each 30 cm long and 1.8 cm high at the crest (Figure 2). A half sine function characterized the stoss side

of the ripple. The lee side of the ripple was characterized by a linear function intersecting the bed at 30 degrees. The size and geometry of these cement ripples were informed by the flume experiments presented in Nelson et al. (2011). The experiments of Nelson et al. (2011) were run in the same flume with the same sediment as the experiments presented herein, assessing fluid



**Table 1.** Experimental Overview

| Run | Distance Downstream ($cm$) | Distance Downstream (*step heights*) | $q_s$ ($\frac{grains}{cm*s}$) | $q_s$ ($cm^2/sec$) |
|---|---|---|---|---|
| 2 | 3 | 1.5 | 28.21 | 0.002 |
| 3 | 8 | 4 | 78.75 | 0.005 |
| 4 | 13 | 6.5 | 135.16 | 0.009 |
| 5 | 18 | 9 | 178.08 | 0.012 |
| 6 | 23 | 11.5 | 282.23 | 0.018 |
| 7 | 26 | 13 | 212.08 | 0.014 |

and sediment dynamics over live bedforms. The present study's cement ripples and sediment discharge were scaled to replicate the live ripples and discharge from Nelson et al. (2011).

One ripple was designated as the sample ripple. This ripple was loaded with live sediment for every experimental run. Mobile sand was well sorted with a median diameter (*D50*) of 0.05cm. The discharge for each run was determined using an inline vortex flow meter and was consistently $\sim$ 0.01 m3/s. The motion of the bedload, illuminated by high-intensity LED lights, was observed with a high-speed camera operating at 250 frames/s. The camera was angled so that the lens was parallel to the sloped bed to minimize distortion due to bed slope and, thus, depth differences. The field of view was approximately 36

cm2 with a 1280 x 1024 pixel resolution. Images were captured at 6 distances along the stoss side of the test ripple (Runs 2-7; Table 3.1). Each run overlapped with the previous run by 1 cm (Figure 2). The flow depth was kept constant at 9.5 cm.

    The experimental procedure was as follows: With the flume off, sand was loaded to the test ripple and screed as best as possible into a planar surface. Once the mobile bed was planar, a Plexiglass sheet with a centimeter ruler grid printed on it was placed on the mobile bed to ensure the sand stayed intact until the beginning of the recording. The flume was then turned on,

and the recirculation rate was raised gradually to a recirculation speed of 17.3 Hz (i.e., re- circulation rate required to attain a discharge of $\sim$ 0.01 $m^3/s$). Once at 17.3 Hz, a Plexiglass window was lowered into the flume to rest on the water surface above the mobile bed to provide optimal image clarity by minimizing distortions from an irregular water surface.

    The camera, mounted on a stable platform, was moved above this window. The camera was focused, and an image was taken of the Plexiglass grid for post-processing scaling. For runs 2 and 3, the Plexiglass grid was pulled, and then the recording of

bedload motions began. For runs 4-7, due to high transport rates, recording began while the Plexiglass grid was still on the bed. The Plexiglass grid was then pulled at the beginning of the recording. For these runs, the first 3 seconds of recording are ignored to account for the Plexiglass grid being pulled and the bed equilibrating.

    Two additional runs without live sediment were conducted to collect fluid velocity data. Streamwise and vertical fluid velocity data were collected using Laser Doppler Velocimetry (LDV). Velocity data were collected for 3 minutes at 15 positions

along the test bedform at 2 cm intervals (Figure 2); measurements were taken at 1mm and 3mm above the bed.





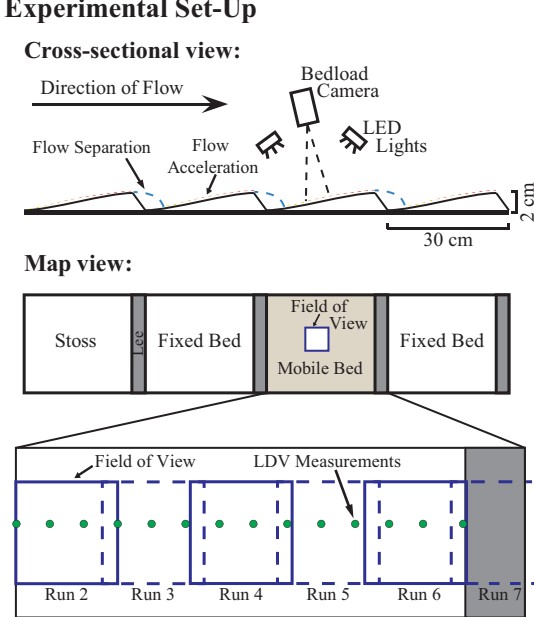

**Figure 2.** Schematic of experimental set-up and measurement locations as well as topography of sample bedform that was loaded with sediment for the experiments.

## 2.2 BEDLOAD TRANSPORT RATE AND PATTERNS

Bedload transport analysis was conducted using bedload images and the open-source software ImageJ. Bedload transport rates were acquired by manually tracking sand particles as they crossed a 6 cm line bisecting the field of view. Transport patterns were determined for each run using the same methods presented in (Leary and Schmeeckle, 2017).

## 2.3 DETERMINING FLOW PATTERNS

LDV fluid velocity data were analyzed as a distribution using basic statistics and as fluid velocity fluctuations using quadrant analysis. LDV yields time-averaged streamwise ($u_x$) and vertical ($u_z$) velocity values. Fluid velocity fluctuations are defined as:

$$u_i' = u_i - \bar{u}_i \tag{1}$$

where $u'$ is the magnitude of the fluid velocity and a given point in time ($u_i$) deviates from the mean ($\bar{u}_i$). The subscript '$i$' denotes the direction of flow (streamwise (x) or vertical (z)). The covariance of streamwise and vertical fluid velocity fluctuations is equal to the Reynolds stress ($-\rho \bar{u'_x} \bar{u'_z}$). Reynolds Stress was calculated for all LDV sampling locations.

Quadrant analysis is a two-dimensional analysis wherein fluid velocity fluctuations, calculated by Equation (1), are paired to produce 4 quadrants that describe the instantaneous movement of the flow (Table 2). Quadrant plots provide a visual representation of the quadrant activity that dominates the flow. Quadrant plots herein include all data points but are binned to illustrate





**Table 2.** Quadrant Overview

| Quadrant | $U'_x$ | $U'_z$ | Contribution to Reynolds Stress |
|:---:|:---:|:---:|:---:|
| 1 | >0 | >0 | − |
| 2 | <0 | >0 | + |
| 3 | <0 | <0 | − |
| 4 | >0 | <0 | + |

the spatial density of the data. Significant quadrant observations were derived from only observations that exceed a threshold (H) value of one standard deviation of the Reynolds stress (Table 3; Lu and Willmarth, 1973).

Flow exuberance, EXFL, was calculated at all LDV sampling locations along the bedform using only significant quadrant observations (observations greater than H). Exuberance describes the shape of the quadrant distribution by using a ratio of the
total Q1 and Q3 events to Q2 and Q4 events (Shaw et al., 1983; Yue et al., 2007; Chapman et al., 2012, 2013). In other words, exuberance is the ratio between positive and negative contributions to the Reynolds stress. We use two methods to calculate flow exuberance: the stress-fraction method (Shaw et al., 1983; Yue et al., 2007) and the time-fraction method (Chapman et al., 2012, 2013). If exuberance is near or equal to | 1 |, there is an even distribution of events in all quadrants, and the resulting quadrant plot is roughly circular. If exuberance values are approaching zero, however, that indicates a dominance of quadrant
2 and quadrant 4 events, and the resulting quadrant plot will be skewed toward those quadrants.

## 3 RESULTS

Mean streamwise fluid velocities increase nonlinearly along most of the bedform, the exception being right at the crest, where mean streamwise fluid velocity decreases slightly (Figure 3). Streamwise fluid velocity data and bedload imagery indicate flow reattachment occurs at approximately 1 step height downstream from the trough (approx. 3 step heights downstream
of flow separation). Mean vertical fluid velocities increase along the bedform up to 10 step heights, where they begin to decrease (Figure 3). The mean vertical fluid velocity is negative at, and just downstream of, flow reattachment and at the crest. Positive vertical fluid velocities dominate the middle portion of the bedform. Patterns in mean streamwise and cross-stream fluid velocities are following observations made in previous studies (e.g., Venditti, 2007; Bennett and Best, 1995; McLean et al., 1994; Nelson and Smith, 1989).

Observed sediment transport increases linearly along the bedform except at the crest, where transport decreases slightly (Figure 3). This pattern of bedload transport is in contrast to results from Leary and Schmeeckle (2017), in which bedload transport downstream of a backward-facing step (i.e., only responding to flow-reattachment) increased nonlinearly (Figure 3). The flow then increased rapidly just downstream of flow reattachment and leveled out with increased distance along the bedform. However, this linear increase in transport with increasing distance along the stoss side of the bedform is necessary
for two-dimensional bedforms to sustain their two-dimensional geometry. Thus, this result is both expected and interesting





in that the difference in flow and bedload transport in the presence of bedforms is precisely the difference required for the self-sustaining migration of bedforms. We will discuss this point in detail in the discussion section.

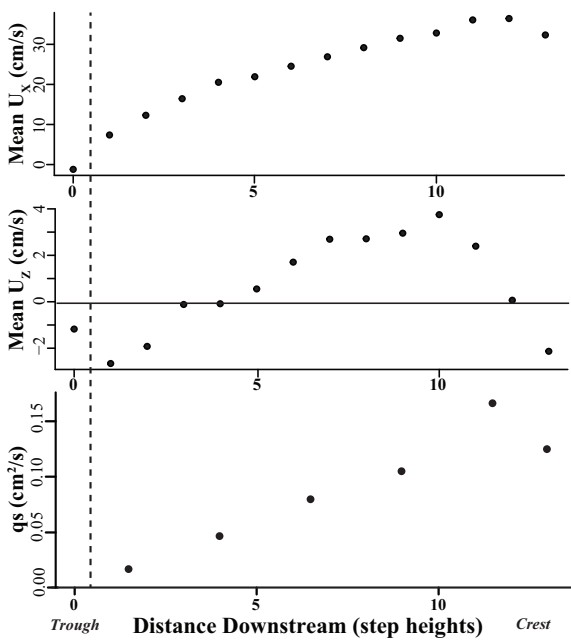

**Figure 3.** Mean streamwise fluid velocities, mean vertical fluid velocities, and observed bedload transport with distance along the bedform. Dashed line indicates the location of flow reattachment (approx. 1 step height along the bedform).

## 3.1 FLUID PATTERNS

Although average streamwise and vertical fluid velocities increase nonlinearly with increased distance along the bedform, the
standard deviations of streamwise and vertical velocity distributions reflect a different pattern (Figures 4A and B). Streamwise and vertical standard deviations peak just downstream of flow reattachment. With increased distance along the bedform, standard deviations of fluid velocities decrease. Near flow reattachment, streamwise and vertical fluid velocities distributions have greater dispersion and higher magnitude fluid fluctuations. In particular, this suggests the potential for large magnitude positive streamwise and negative vertical fluid velocity fluctuations. Fluctuations of this type have been observed to be significant
factors in splat events (Stoesser et al., 2008; Schmeeckle, 2015; Leary and Schmeeckle, 2017).

This pattern of standard deviations with increased distance along the bedform is congruent with increased Reynolds stresses in the region proximal to flow reattachment (Figure 4). Reynolds stress is a measure of the covariance of fluid fluctuations in the streamwise and vertical directions. Reynolds stress decreases in magnitude with increasing distance along the bedform, except at the crest, where it is slightly higher than immediately upstream (Figure 4C). This pattern of Reynolds stress is in
agreement with previous studies' findings (e.g., Bennett and Best, 1995; Venditti and Bennett, 2000; Robert and Uhlman, 2001; Venditti and Bauer, 2005; Fernandez et al., 2006), wherein decreasing Reynolds stress along the stoss side is due to the



Earth **Surface**
**Dynamics**
Discussions

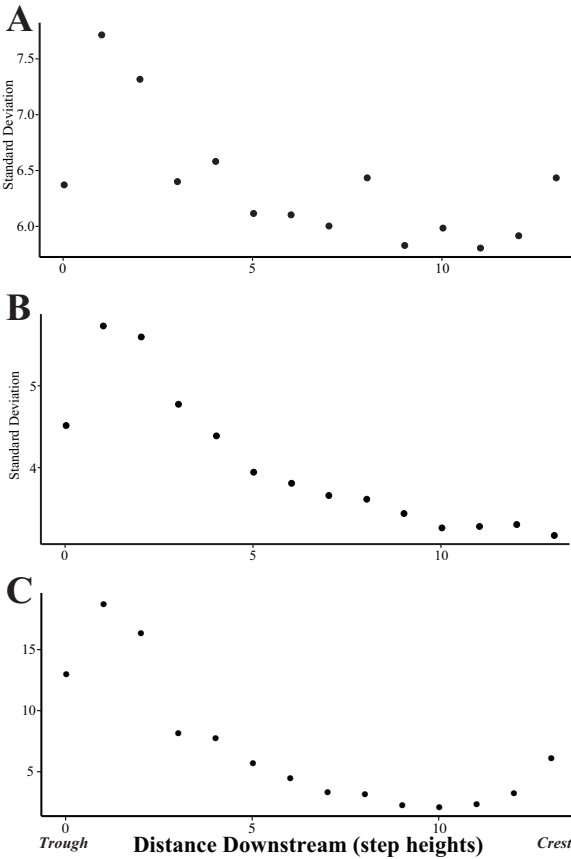

**Figure 4.** Standard deviation of streamwise (A) and vertical (B) fluid velocity distributions. Higher standard deviations near flow reattachment indicate the potential for high magnitude fluid fluctuations. (C) Reynold Stress with distance along the bedform. Reynold stress was calculated from LDV velocity data collected 3mm above bed.

development of the internal boundary layer. Measurements near flow reattachment and on the lee side are in the wake region of flow reattachment and therefore have elevated Reynolds stress values. Conversely, measurements made along the stoss side of the bedform are within the internal boundary layer and therefore have greatly reduced Reynold stress values.

Quadrant analysis conducted at 2, 7, and 12 step heights is also congruent with the above statistical analysis of the flow (Figure 5; Table 3). At 2 step heights (proximal to flow reattachment), we see the dominance of quadrant 2 and 4 events, which are composed of high magnitude streamwise and vertical fluctuations. At 7 and 12 step heights, however, all quadrants are roughly equally represented. Additionally, whereas at 2 step heights, the data is oriented towards quadrants 2 and 4, quadrant plots at 7 and 12 step heights are oriented elongate in the $U'_x$-direction and narrower in the $U'_z$-direction. This change in pattern

with increased distance along the bedform indicates that at distances medial and distal to reattachment, the fluid is experiencing more significant magnitude fluctuations in the streamwise direction compared to the vertical direction.



**Table 3.** Summary of Significant Quadrant Events and Exuberance (EXFL)

| Distance (step heights) | Distance (cm) | Q1 | Q2 | Q3 | Q4 | Total | % Q1 | % Q2 | % Q3 | % Q4 | Q2:Q4 | EXFL (Time-Fraction) | EXFL (Stress-Fraction) |
|---|---|---|---|---|---|---|---|---|---|---|---|---|---|
| 0 | 0 | 77 | 534 | 70 | 610 | 1291 | 6 | 41 | 5 | 47 | 0.88 | 0.13 | -0.09 |
| 1 | 2 | 98 | 756 | 111 | 699 | 1664 | 6 | 45 | 7 | 42 | 1.08 | 0.14 | -0.10 |
| 2 | 4 | 116 | 688 | 108 | 719 | 1631 | 7 | 42 | 7 | 44 | 0.96 | 0.16 | -0.13 |
| 3 | 6 | 184 | 518 | 188 | 573 | 1463 | 13 | 35 | 13 | 39 | 0.90 | 0.34 | -0.31 |
| 4 | 8 | 209 | 522 | 160 | 611 | 1502 | 14 | 35 | 11 | 41 | 0.85 | 0.33 | -0.28 |
| 5 | 10 | 220 | 533 | 149 | 541 | 1443 | 15 | 37 | 10 | 37 | 0.99 | 0.34 | -0.31 |
| 6 | 12 | 259 | 516 | 170 | 514 | 1459 | 18 | 35 | 12 | 35 | 1.00 | 0.42 | -0.38 |
| 7 | 14 | 270 | 479 | 185 | 483 | 1417 | 19 | 34 | 13 | 34 | 0.99 | 0.47 | -0.46 |
| 8 | 16 | 325 | 457 | 214 | 475 | 1471 | 22 | 31 | 15 | 32 | 0.96 | 0.58 | -0.52 |
| 9 | 18 | 328 | 443 | 235 | 449 | 1455 | 23 | 30 | 16 | 31 | 0.99 | 0.63 | -0.59 |
| 10 | 20 | 330 | 453 | 214 | 429 | 1426 | 23 | 32 | 15 | 30 | 1.06 | 0.62 | -0.57 |
| 11 | 22 | 287 | 427 | 220 | 415 | 1349 | 21 | 32 | 16 | 31 | 1.03 | 0.60 | -0.57 |
| 12 | 24 | 237 | 495 | 188 | 497 | 1417 | 17 | 35 | 13 | 35 | 1.00 | 0.43 | -0.39 |
| 13.27 | 26.54 | 166 | 625 | 117 | 583 | 1491 | 11 | 42 | 8 | 39 | 1.07 | 0.23 | -0.20 |





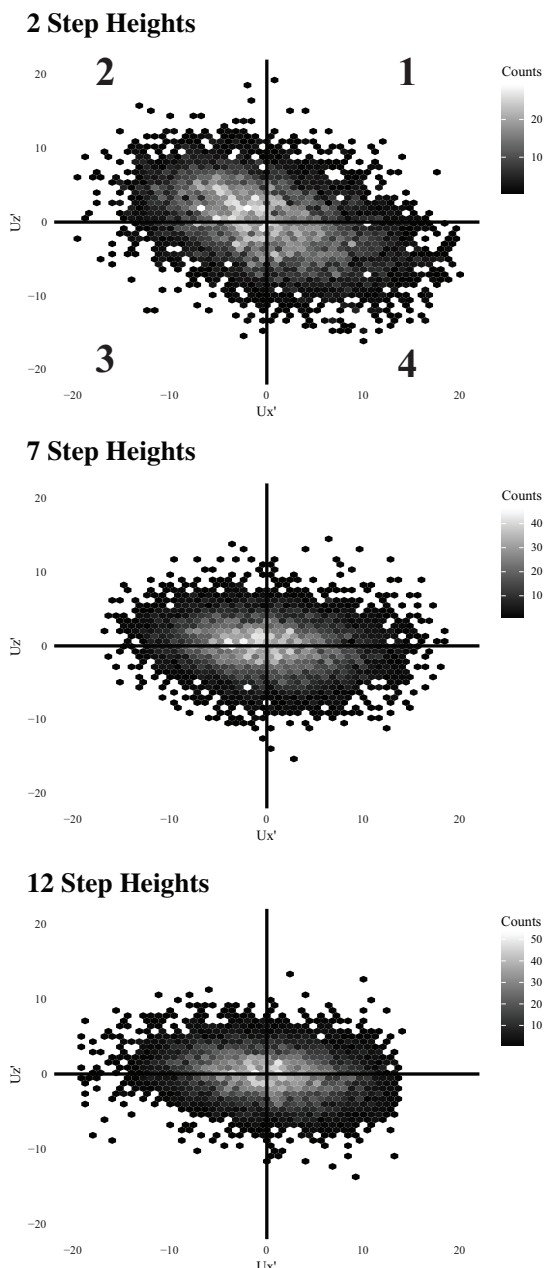

**Figure 5.** Quadrant analysis at three different distances along the bedform. Data are hexagonally binned (nbins=50 in each direction) to display varying density of data.

Flow exuberance also captures this change in quadrant distribution with increasing distance along the bedform (Figure 6). Both time- and stress-fraction exuberance are nearest to 0 in the region near flow reattachment, indicating that the region is





mainly dominated by quadrant 2 and 4 events. With increasing distance along the bedform, however, exuberance increases
towards | 1 |, indicating an increase in the frequency of quadrant 1 and 4 events. Chapman et al. (2012) identified this "exuberance effect" over coastal eolian dunes. They observed that Reynolds stress increased when the time-fraction exuberance was low (near the toe and lower stoss region). This is expected as low time-fraction exuberance indicates dominance of Quadrant 2 and 4 events that contribute positively to the Reynolds stress.

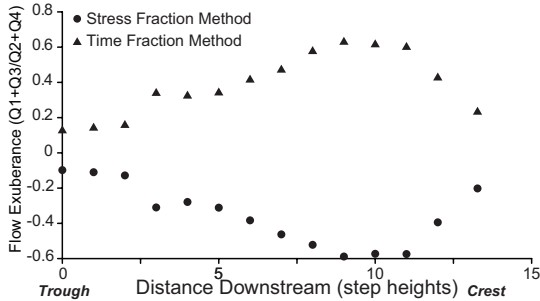

**Figure 6.** Time- and stress-fraction exuberance at all 14 LDV sampling locations along the bedform. In the region proximal to reattachment, Quadrant 2 and 4 events are dominant as indicated by exuberance estiamtes near 0. Farther along the ripple, exuberance estimates diverge from 0 as Quadrant 1 and 3 become more prevalent.

## 3.2    PATTERNS OF BEDLOAD TRANSPORT

Streamwise and cross-stream bedload transport time series data at 1.5, 6.5, and 11.5 step heights show similar patterns to those observed by Leary and Schmeeckle (2017). Transport is intermittent at all distances along the bedform (Figure 7A; Supplemental Movies 1-3). Near flow reattachment (1.5 step heights) streamwise and cross-stream transport is of similar magnitudes. With increased distance along the bedform, however, streamwise transport increasingly outweighs cross-stream transport (Figure 7A). The direction of transport data from manual particle tracking supports these observations (Figure 7B).
Near flow reattachment, transport occurs in a wide range of directions (+90 to -90 degrees). With increased distance along the bedform, the direction of transport narrows to just the streamwise direction (within the range of +22.5 to -22.5 degrees).

     Near flow reattachment, transport is much more localized than further downstream (Figure 8). At 1.5 step heights (Figure 8A), almost all the transport observed is initiated in the upper left-hand corner of the field of view at the beginning of the transport event. At 7 and 12 step heights (Figure 8B and 8C), however, transport is initiated and occurs throughout the field
of view and throughout the transport event. Leary and Schmeeckle (2017) contributed these localized, intermittent, high-magnitude, multidirectional transport events, observed near flow reattachment, to bedload patterns associated with splat events. The localized initiation and radial transport pattern observed in Figure 8A and Supplemental Movie 1 reflect the pattern expected of a splat event (Perot and Moin, 1995; Stoesser et al., 2008; Schmeeckle, 2015; Leary and Schmeeckle, 2017).





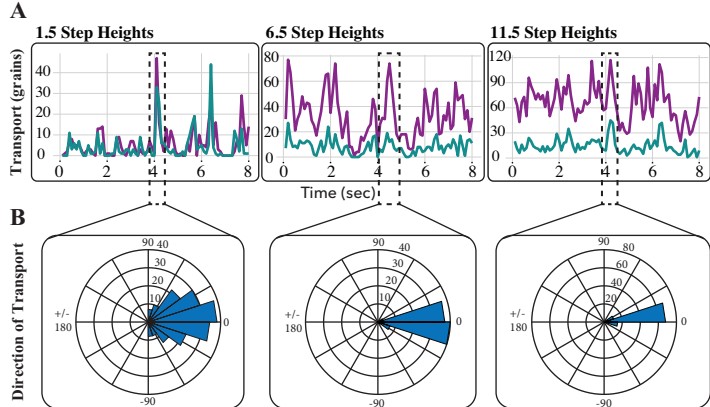

**Figure 7.** (A) Times series of streamwise (purple) and cross-stream (green) bedload transport with increasing distance along the bedform. Streamwise transport is characterized particles being transported within the directional range of -22.5 to 22.5 degrees. Cross-stream transport is characterized by particles moving in the directional range of 22.5 to 90 degrees or -22.5 to -90 degrees. Anything higher that 90 degrees is considered upstream transport. (B) Rose diagrams indicating the direction of transport form manual particle tracking of the transport event outline is the black, dashed box for each distance along the bedform. Direction of transport becomes more dominated by streamwise transport with increased distance along the bedform. Near flow reattachment, sediment has a wide range of directions in which it is transported.

## 4 DISCUSSION

### 4.1 LINEAR PATTERN OF TRANSPORT RATES

The pattern of sediment transport rates downstream of flow reattachment presented herein contrasts with Leary and Schmeeckle (2017). When flow reacceleration is present in addition to flow separation/reattachment, bedload transport rates increase linearly with increased distance along the bedform (Figure 3). A linear increase in transport is necessary for bedforms to retain a two-dimensional geometry while translating downstream. With a greater developed flow and acceleration, the greater the sediment transport occurrence for the migration of the bedform downstream (Terwisscha van Scheltinga et al., 2021). Tsubaki et al. (2018) demonstrated that two-dimensional bedforms display a simultaneous sequence of transport events over a broad area. This specific particle movement was seen across adjacent bedform crests and troughs, producing the same transport patterns and maintaining coherent two-dimensional structures due to the uniform velocity- this could contribute to two-dimensional bedforms maintaining their shape.

Consider conservation of mass of the bed in which there is no exchange of suspended sediment with the bed:

$$-\frac{\delta z}{\delta t} = \left(\frac{\delta q_s}{\delta x} - \frac{\delta q_s}{\delta y}\right)\frac{1}{1 - \lambda_p} \tag{2}$$

Where $q_s$ is the sediment transport rate in the streamwise direction, $\delta z/\delta t$ is the erosion rate, and $\lambda_p$ is the porosity of the sediment. Lets first assume that $\delta q_s/\delta y = 0$. If $q_s$ increases with respect to $x$, erosion occurs. If $q_s$ decreases with respect to $x$, deposition occurs. This is in agreement with the classical formulation that as bedforms migrate, sediment is eroded along

Earth **Surface**
Dynamics
Discussions

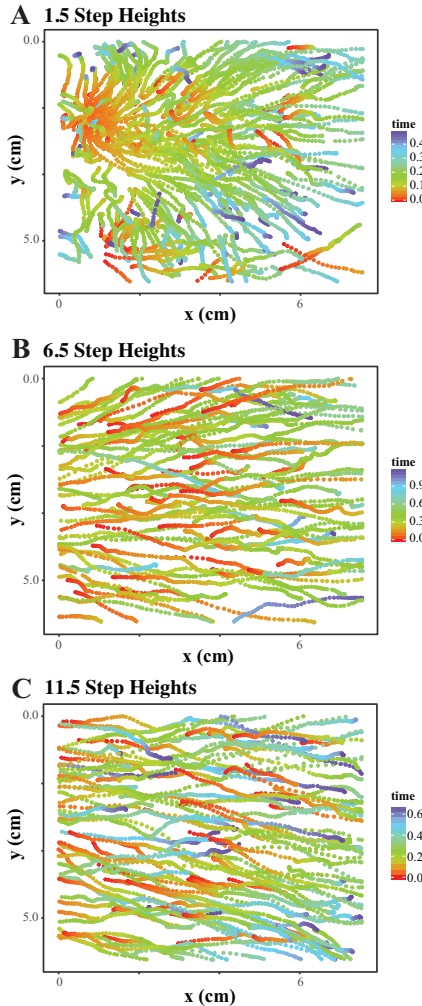

**Figure 8.** Sand grains in transport through time at 1.5, 6.5, 11.5 step heights. Grains were tracked during the transport events outlined in figure 7A. At 1.5 step heights, a majority of grains in transport during the transport event are entrained at a localized position in the upper right hand corner of the field of view at the beginning of the transport event and continue to be in transport until the end of the transport event. At 6.5 and 11.5 step heights, however, particles are being entrained and transport at all location and times.

the stoss side of the bedform (where sediment transport rates increases due to increasing mean streamwise fluid velocities) and deposited on the lee side (where sediment transport rate decreases due to flow separation).

There remains a conundrum, however. If erosion is occurring along the stoss side of the bedform, why are bedforms long-lived features? Why do they not simply erode away? The pattern of bedload transport presented in this study suggests that deposition is initiated at the crest of the bedfrom (where $q_s$ begins to decline; Figure 3) and continues over the lee side of the bedform. This pattern of peak $q_s$ located upstream of the crest is integral to maintaining bedforms because it initiates deposition at the crest rather than continuing to erode the bedform away. Why does a decrease in $q_s$ occur at the crest? The crest represents





the region of the bedform in which the zone of fluid acceleration transitions to flow separation. At this transition, near-bed fluid velocities decrease (Figure 3). Additionally, flow separation does not occur at a fix point in space and time. This variability creates a flow separation "zone" at the crest that is characterized by decreases in near-bed fluid velocities (Figure 3) resulting

in a decrease in $q_s$ at the crest.

The rate of change of $q_s$ along the stoss side of the bedform also has important implications for the geometric evolution of bedforms. For erosion to occur on the stoss side of the bedform only an increasing pattern of transport is necessary (i.e. it is not necessary for transport rate to increase linearly). A linear increase in transport rate is necessary, however, to maintain a constant erosion rate and therefore the two-dimensionality of the bedform. Substituting a linear equation for $q_s$ into equation

(2) results in:

$$-\frac{\delta z}{\delta t} = \frac{\delta(ax+b)}{\delta x}\frac{1}{1-\lambda_p} \tag{3}$$

where $a$ and $b$ are constants. Solving the derivative for change in sediment transport rate with respect to $x$ thus produces a constant rate of erosion independent of distance along the bedform:

$$-\frac{\delta z}{\delta t} = \frac{a}{1-\lambda_p} \tag{4}$$

In this case of a linear increase in sediment transport rate, in which there is no cross-stream variability ($\delta q_s/\delta y = 0$), the bedform will erode an equal amount at all distances along the stoss side and, assuming all that sediment is then deposited on the lee side (i.e. no suspension), therefore retain a two-dimensional geometry (Figure 9A). Any nonlinear increase in sediment transport rate could result in deformation (i.e. when the sum of all changes in elevation of the bed does not equal zero; McElroy and Mohrig (2009)) and potentially cause a shift to a more three-dimensional geometry, especially if variability in the cross-

stream direction exists ($\delta q_s/\delta y \neq 0$).

Venditti et al. (2005) reported the development and importance of 'crest defects' in the transition from two-dimensional to three-dimensional bedforms. Small excesses or deficiencies of sediment at the crest line cause these crest defect features (Venditti et al., 2005). As time elapsed and flow conditions remained constant, Venditti et al. (2005) observed that the field of bedforms (originally two-dimensional) became overwhelmed by crest defect features and transitioned to a field of three-

dimensional bedforms.

Based on results from this study, we hypothesize that crest defects could be caused by a spatially non-uniform increase in transport rates along the bedform in the cross-stream direction (i.e. linear increase in some regions, nonlinear increase in other regions). For example, if sediment transport rates increase exponentially (i.e. $q_s = x^a$; where $a > 1$), the erosion rate will increase along the stoss side causing a deficiency in sediment near the crest where the erosion rate is highest (Figure 9C). In

contrast, if sediment transport rates increase logarithmically (i.e. $q_s = log(x)$), erosion rates will decrease with distance along the stoss side resulting in an excess of sediment near the crest where erosion rate is lowest (Figure 9B). The spatiotemporal changes in bedload transport rate over bedforms need to be examined in more complex conditions than that of a fixed, two-dimensional ripple (as presented in this study) in order to determine the validity of the above hypotheses.





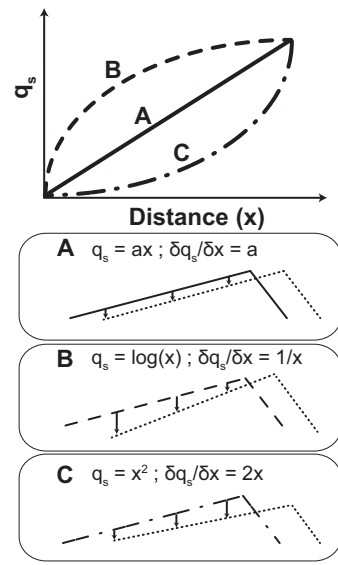

**Figure 9.** Schematic of different patterns of erosion along the stoss side of a bedform based on linear (A), logarithmic (B), and exponential (C) increases in bedload transport.

## 4.2 THE DYNAMICS OF SPLAT EVENTS

In addition to changes in the overall magnitude of transport occurring along the stoss side of the bedform, changes in the pattern of transport and turbulent structures also occur. In the region just downstream of flow reattachment, the fluid is dominated by large magnitude streamwise and vertical fluid fluctuations that take the form of either quadrant 2 or 4 events. The dominance of these events decreases with increased distance along the stoss side of the bedform. Notably, quadrant 4 events are integral to splat events (Schmeeckle, 2015; Leary and Schmeeckle, 2017), and the increase in these events near flow-reattachment

indicates that splat events may be occurring in the region. Bedload transport time series and manual particle tracking indicate that in this zone just downstream of flow reattachment, transport is localized, intermittent, high-magnitude, and multidirectional—the same characteristics previously attributed to particles transported by splat events (Leary and Schmeeckle 2017). These results indicate that splat events still play a significant role in the transport pattern in the zone immediately downstream of flow reattachment, even when flow reacceleration is present. The majority of transport occurring at 1.5 step heights is the

result of a splat event. We can use particle tracking data to assess the transport characteristics of splat events. For the splat event observed at 1.5 step heights, the length of transport and particle velocity are investigated in relation to the transport direction.

Although splat events initiate transport in a radial pattern, transport velocity (both mean and instantaneous) and transport length (both cumulative and instantaneous) vary with the direction of transport (Figure 10). 'Instantaneous' refers to transport dynamics (length, velocity, and direction) at each time step. Mean velocity is the average speed the particle moves throughout

active transport. Cumulative transport length is the distance the particle travels the entire time it is in motion. Instantaneous and cumulative data show that particles moving in the streamwise direction have a much larger velocity distribution and transport





length. At a maximum, particles traveling in the streamwise direction have a velocity and transport length approximately double that of a particle moving in a cross-stream direction. This indicates that splat events do not transport particles equally in all directions. Despite this, splat events do actively transport sediment in the cross-stream direction, indicating that cross-stream

transport may play a more active role in bedload transport over bedforms than previously thought.

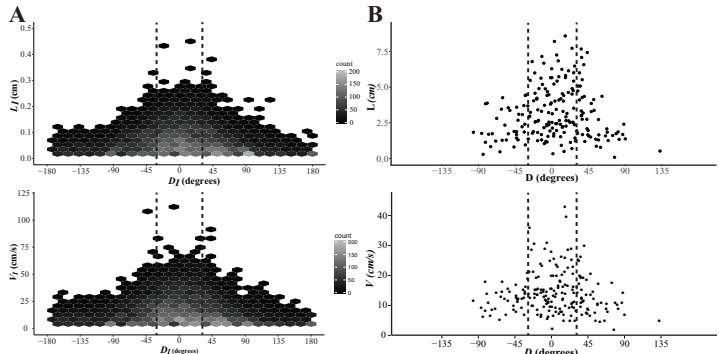

**Figure 10.** Instantaneous (A) and cumulative (B) transport characteristics or a splat event. Vertical dashed lines indicate +/130 degrees, transport between which is classified as streamwise. (A) Instantaneous length of transport (LI) and instantaneous transport velocity (VI) referenced to direction of transport. The data is binned hexagonally to illustrate the density of the data (nbins = 25; n = 11,091). (B) Cumulative track length (total distance particle travels) and mean particle velocity during over the entire period of transport.

It is also worth noting that at all locations where particles were tracked, particles were not observed saltating in a classical sense. That is to say, with these coarser sediments, particles are not observed being ejected into the flow or saltating with large hop distances. Instead, particles appear to almost trundle along the surface of the bedform. This agrees with Fathel et al. (2015), where streamwise and cross-stream particle motions over a flatbed exhibited predominantly small hop distances.

Radice (2021) experimental work determined that the sediment concentration appears to be the main factor in the transport of sediment along a dune. This could affect the transport values in the streamwise and cross-stream direction of the flow field. Radice (2021) noted that the particle migration primarily held a one-dimensional flow in the downstream direction, with cross-stream transport only being reported in the trough, and few particles at that. Depending on the strength of the flow and the concentration of sediment in the bedload, does a higher bedload concentration create more splat events therefore initiating a

higher particle movement in the cross-stream direction?

There remain some biases in this method of particle tracking. The first is that particles that are moving slowly are much easier to track. Although an effort was made to track particles randomly regardless of speed, this unintended bias is potentially still present. For this reason, particle velocities may be greater than those presented in Figure 10. Secondly, the length of transport is, of course, biased by the field of view size. Once a particle leaves the field of view, its track is terminated, but it may

continue to be transported. Therefore, the transport lengths reported in Figure 10 should be considered minimum estimates. Lastly, small particle displacements, in which particles are transported on very short timescales, are often not taken into account during manual particle tracking (Fathel et al., 2015; Finn et al., 2016). Fathel et al. (2015; 2016) found that these small particle



displacements dominate bedload motions over a flatbed. The bedload tracking analysis conducted herein did not expressly address this, so the lower end of transport length and velocity distributions may not be represented.

### 4.3 THE IMPORTANCE OF SPLAT EVENTS

The dynamics of splat events not only inform our understanding of the importance of cross-stream transport proximal to flow reattachment, but they also potentially provide insight into the three-dimensionality of bedforms. For example, Rubin and Ikeda (1990) and Rubin and Hunter (1987) demonstrated that bedform alignment in multidirectional flows depends on the maximum gross bedform normal transport, which is dictated by the resultant vector of two flow vectors. Although these studies did not investigate flows with more than two flow vectors, the concept of shifting dominant transport directions depending on flow geometry and, by extension, bedform geometry is intriguing.

Building on the experiments of Allen (1966), Venditti (2007) investigated the flow patterns over non-planform dune geometries and found that flow over a lobe shape tended to converge downstream. In contrast, flow over a saddle shape would diverge. Splat events may become concentrated in these regions of flow convergence or divergence, potentially shifting the direction of maximum gross bedform normal transport. The convergence and divergence of flow over lobe/saddle features could potentially cause an along-dune variability in the intensity of splat events. Along-dune variability in the intensity of splat events could produce a gradient of sediment transport with respect to y (i.e., $\delta qs/\delta y * 0$). As noted in the above discussion, variability in transport rates in the cross-stream direction would enable deformation of the downstream crest and induce bedform three-dimensionality. Considering the results presented herein, we suggest two potential mechanisms that drive the transition from two-dimensional to three-dimensional bedform geometries: (1) splat events near flow reattachment and (2) localized, nonlinear increases in bedload transport rates along the stoss side of the bedform. These two processes may be genetically linked, and we suggest that (1) could drive (2).

## 5 Conclusions

We assess the effects of flow separation and flow reacceleration on sub-bedform bedload transport dynamics using high-resolution velocity (ADV) and bedload data (high-speed imagery; manual tracking techniques) from flume experiments of bedload transport over fixed, 2D ripples. Results reported herein show that:

1. Mean bedload transport rates increase linearly with distance along the stoss side of bedforms.

2. Continuity, velocity, and direction of bedload transport vary significantly with in- creased distance along the stoss side of the bedform.

3. Splat events continue to play a pivotal role in bedload transport near flow reattachment when flow reacceleration is present.

The existence and importance of splat events are congruent with previous studies that lacked the detailed bedload tracking analysis included in this study but that recognized the importance of quadrant 1 and 4 events in the entrainment of bedload near

flow reattachment [Bennett and Best, 1995; McLean et al., 1994a]. Results reported herein and by Leary and Schmeeckle (2017)
and Schmeeckle (2015) indicate that splat events are (1) the primary mechanisms entraining sediment near flow reattachment,
(2) comprised of quadrant 1 and 4 events (or the octant sequence  1 4 4  in the case of Leary and Schmeeckle (2017)), and
(3) entrain sediment in both the streamwise and cross-stream directions. Although splat events transport sediment at greater
velocities and greater distances in the streamwise direction, their transport dynamics in the cross-stream direction remain
significant. Further work needs to be done investigating the spatiotemporal patterns of transport rates over live bedforms, and
the bedform-scale effect splat events have on along-dune transport.

*Data availability.* All data presented in this article are available at https://doi.org/10.5281/zenodo.7552715. Data are also available upon
request to kate.leary@nmt.edu).


*Author contributions.* KCPL designed and performed the experiments and processed the resulting data. All authors were responsible for
developing the critical ideas present in the manuscript as well as approving the final submission and editing text/figures.

*Competing interests.* There are no competing interests.

*Acknowledgements.* This research was supported by a National Science Foundation research grants (award #1734752) awarded to Mark
Schmeeckle. I am very thankful to the reviewers for their thoughtful and constructive reviews.



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
