# Peer review of "Spatiotemporal Bedload Transport Patterns Over Two-Dimensional Bedforms"

_Earth Surface Dynamics, 2023_

## Author Comment (AC1)

*We thank the reviewer for their thoughtful and thorough review of the manuscript. Below are our responses to major and minor comments in green italic.*

**Referee #1 Comments:**

This study examines hydrodynamics and sediment transport over a sequence of laboratory bedforms. The key message is that the specific flow patterns driving sediment transport vary across bedforms. On the stoss side of bedforms, Reynolds stresses decrease as sediment transport increases. Maximum transport is reached just before bedform crests, a transport pattern which would act to maintain downstream migrating bedforms. At bedform crests, the flow detaches before it eventually reattaches on the lee side. Reattachment zone flow involves a higher relative frequency of Q2 and Q4 events, diagnostic of the "splat events" identified in the authors' earlier publications on the flow behind a step. Sediment transport in the reattachment zone is multidirectional and intermittent, consistent with the hypothesis that splat events dominate sediment transport there. These results summarize the linkages between turbulent hydrodynamics and sediment transport over bedforms and serve as a useful contribution, so the paper is therefore in my opinion appropriate for ESurf after some revisions aimed at clarifying its presentation and improving its readability.

**Main comments:**

The literature review could engage more directly with the studies it cites near L24 which relate turbulent hydrodynamics and sediment transport. This would better place the present study in context. In particular, statements of what each study (or sets of studies) did and didn't do could better indicate the knowledge gap the authors examine. The text would ideally give careful attention to the findings of studies which have jointly measured hydrodynamics and transport, e.g. Nelson et al (1995) and related studies extending to the present day. In addition, the literature review and discussion do not engage much with the numerical simulation literature, which has been an extremely successful method to study the interaction between turbulence and sediment transport. It may be useful to incorporate the wider numerical literature on the interaction of turbulent hydrodynamics and sediment transport over bedforms into the discussion (e.g., other studies relatex to Schmeeckle 2015 and surely more recent work).

> *Rather than to add a broad discussion of numerical simulations involving flow, sediment transport and bedforms as suggested by the review, we have chosen to pick a rather few recent studies that bear on the knowledge gap driving this study. In particular, we focus on the recent work of Kidanemariam et al. (2022) which focuses on their DNS-DEM model. In accordance, we have added the following paragraph to the introduction that addresses recent numerical studies investigating bedload transport over bedforms.*

> *Many successful numerical models of flow and sediment transport over bedforms have involved the Exner equation combined with an algebraic equation for the relationship between bed stress and sediment flux\citep[e.g.,][]{khosronejad2014numerical,chou2010model, zgheib2018direct}. Some of the more successful models have used a slightly more complicated formulation involving the pickup and deposition of particles, rather than a simple formula between stress and flux \citep[e.g.,][]{giri2006numerical, nabi2013detailed}.*

*Formulas like this allow for a lag between stress and flux. However, the experiments of Nelson et al. (\citeyear{nelson1995}) and Leary and Schmeeckle (\citeyear{Leary_Schmeeckle_2017}) downstream of a negative step suggest that the relation between stress and flux cannot be well modeled with a simple relation involving either a pickup- deposition formulation, or equivalently, a saturation length formulation. In the experiments of Leary and Schmeeckle (2017), even where stress is apparently negligible near the point of reattchment, downstream sediment flux is substantial. This work and the LES-DEM model of Schmeeckle 2015 show that the temporal variation of sediment flux is large relative to either that over a flat bed or near the crest of a bedform. Kidanemariam et al. (\citeyear{kidanemariam2017formation}) conducted numerical experiments using a DNS of fluid coupled to a DEM of particle motion; small bedforms emerged. Kidanemariam et al. (\citeyear{kidanemariam2022open}), similar to the results of Leary and Schmeeckle (2017) discussed above found that the relationship between bed stress and sediment did not follow a simple algebraic relationship that could be resolved. In fact, Kidanemariam et al. (2022) found that the sediment flux could vary by nearly an order of magnitude for the same stress, depending on the location over the bedform. Turbulence-resolving models coupled to DEM's of particle motion have found that sediment mobility can increase greatly with increasing intensity of near-bed vortical structures \citep{schmeeckle2014numerical,schmeeckle2015role,mazzuoli2020interface} . It is unlikely that an accurate model of stress and sediment flux over bedforms can be constructed without a detailed description of the spatio-temporal pattern of sediment flux.*

The paper might benefit from a graphic summarizing the key relationships observed by the authors between sediment transport and hydrodynamics over bedforms. Currently there is a repurposed diagram (with a copy-pasted caption) from the authors' earlier JGR paper indicating what a splat event is. In my opinion ,the key findings of the paper would be easier to understand with a summary diagram, tailored to the paper, showing (a) the stoss zone of increasing sediment transport and decreasing Reynolds stresses, with a shift in behavior just before the crest; (b) the detachment zone and its associated fluid dynamics; and (c) the reattachment zone and associated prevalence of splat events. Maybe the earlier JGR figure could feature as an inset to show the reader what a splat event is. The present work deserves an original figure tailored to its main message.

- *We have redrafted figure 1 to incorporate the reviewer's comments. We thank the reviewer for catching the mistake in our caption, which is now entirely re-written.*

A highlighted point in the conclusion of the paper is that "Continuity, velocity, and direction of bedload transport vary significantly with increased distance along the stoss side of the bedform." The presented analyses make a convincing point that the direction and velocity vary significantly, but the paper could be more convincing about the point on "continuity" (which I would rather call "intermittency" in deference to standard terminology in the literature). To make the point about intermittency, the authors might plot the variance of the interarrival time of particles to the reference surface used to evaluate $q_s$ as a function along the bedform, or possibly the timeseries of particle arrivals (rather than the flux) -- one should see additional burstiness in the splat-dominated transport zone.

- *We have changed the phrasing to be "intermittency" rather than "continuity." To observe this intermittency and how it changes along the bedform, we refer the reviewer to figure 7A where we have plotted our transport of grains timeseries. This figure shows that near flow reattachment, transport is much more intermittent (has many time periods of zero or near zero transport in the streamwise and cross-stream directions) compared to further downstream along the bedform. We have added more concrete discussion of this point to the first paragraph of section 3.2.*

Finally, I believe the discussion around the Exner equation near Eq. 2 needs attention. To start with, the statement of the Exner equation is incorrect (missing minus sign), and the assertion that "a linear increase in transport is necessary for bedforms to retain a two-dimensional geometry while translating downstream" seems inconsistent with successful nonlinear formulations of bedform dynamics based on the Exner equation (Jerolmack and Mohrig, 2005). A rewrite of this section fixing the conceptual errors and linking more carefully to the extensive literature relating bedform dynamics to turbulent hydrodynamics would benefit the paper.

- *We have modified section 4.1 to be clearer. We have removed the assertion that, "a linear increase in transport is necessary for bedforms to retain a two-dimensional geometry while translating downstream" for the reasons noted by the reviewer. The point we are trying to make is that a linear increase in transport rates is the only scenario in which erosion on the stoss side is equal at all locations. Yes, two-dimensional bedforms can be formed from other, non-linear patterns of sediment transport but they do not yield a constant erosion rate along the entirety of the bedform. With hypothesize that temporal and spatial changes in transport pattern (e.g. linear to logarithmic, or linear to algebraic through time and space) could cause conditions prone to generate crest defects as seen by Venditti et al. (2005).*

**Minor comments:**

- L7 - transport directions (plural)
    - *We have made this correction.*
- L9 - downstream and vertical fluid velocities
    - *We have made this correction.*
- 18 - It may be useful to define the "sub-bedform" scale more carefully
    - *We have added a definition of "sub-bedform" scale*
- 19 - "our understanding of how bedforms evolve in three dimensions" would be more concrete
    - *We have rephrased as suggested.*
- 29 - This seems not exactly correct. Ashley et al did this, and if I recall, Nelson et al (1995) have as well. Heyman et al (2016) JGR:ES also did this, in a way. It would be more correct to say "With few exceptions (e.g., Ashley et al 202X, Nelson et al 199X), previous work has not accounted for the amount of sediment being ...

- *We have adjusted the phrasing to reflect these citations.*

- I believe all citations of the type "Bennett and Best 1995 showed that" will need parentheses around the year.

  - *We have fixed this typo.*

- 35 - Notably, they observed that quadrant 4 events near flow reattachment contribute significantly to . . .

  - *We have made this adjustment.*

- Figure 1: This figure could be improved. The caption (which is copy-pasted from the JGR paper) also has a typo (which is also in the JGR paper) -- characterize..d

  - *We have redrafted this figure and rewritten the caption.*

- 75 Exponents in units should be rendered as superscripts. Also D50 should be rendered (in my opinion) as $D_{50}$.

  - *We have adjusted this notation as suggested.*

- 89 Suggest "removed" instead of "pulled" for clarity

  - *We have made this adjustment.*

- 105: $u'$ is rendered incorrectly as u'

  - *We have adjusted this notation.*

- 107: The reynolds stress is defined as $-\rho \overline{u_x'u_y'}$, not $-\rho \overline{u_x'}\overline{u_y'}$. I assume this is just a typo? If not, the calculations require attention as these are distinctly different quantities.

  - *The reviewer is correct, this was just a typo and does not affect any of the calculations*

- Table 2: change in notation from $u_x'$ to $U_x'$. Also see L159.

  - *We have changed the notation to be consistent.*

- Section 2.2 - How long did you average through time to compute the mean sediment transport rate, and how did this affect your results? See Singh et al (2009), Ancey & Pascal (2020): it is now well established that mean transport rates depend on the averaging time, precisely because transport is intermittent. Additionally you should mention how long your sediment transport analyses lasted. It seems the answer is 8 sec which is exceedingly short - a limitation that deserves mention.

  - *Bedload transport rates were calculated over an 8-second interval. We have added this information to section 2.2. Although 8 seconds seems short, bedload images were taken at a frequency of 250 frames per second resulting in 2000 images at 0.004 second increments. We believe this is sufficient data to accurately establish sediment transport rates and patterns.*

- 119: I assume you actually mean if the absolute value of exuberance is equal to 1, not what you wrote. Suggest "If exuberance is near $\pm 1$, ..."

  - *We have adjusted the notation as suggested.*

- 129: Just style - but consider switching "fluid velocities are following observations" to "fluid velocities follow observations"; similarly at 131, "transport is in contrast" becomes "transport contrasts" and many other locations in the paper - concise and active

    o *We have adjusted the phrasing accordingly.*

- Fig 3: subscript not rendered properly in $q_s$ y axis label

    o *We have fixed the notation accordingly.*

- Fig 4 needs units

    o *We have added units to figure 4.*

- Eq. 2: $\delta$ is traditionally reserved for variational derivatives or finite increments. The Exner equation requires $\partial$. Also, both spatial derivatives should have the same sign. Finally, $\partial z/\partial t$ is only an erosion rate if $\nabla q > 0$. Otherwise, it is a deposition rate.

    o *We have adjusted the notation accordingly.*

- Fig 6: "diverge" suggests an explosion toward infinity. suggest "depart". Also a typo - - "estiamtes" - suggest to spell check the entire document.

    o *We have changed the phrasing and corrected the typo.*

- Fig 7: "characterized particles" (typo)

    o *We have fixed this typo.*

- L171: As I mentioned before, intermittency implies the mean transport rate depends on observation time, so there is a need to describe the definition of mean rate used in Figs 3 and 7 with citation to the relevant papers on the topic.

    o *We have added more information on how we calculated mean transport rates to section 2.2. Mean transport rates (figure 3) are defined as the average sediment transported over an 8 second observation window (2,000 frames). The time series of sediment transport (figure 7) were generated by tracking and counting grains as they passed over a line bisecting the field of view in the streamwise and cross-stream directions every 25 frames (0.1 second) for the entirety of the 8 second observation window.*

- Fig 8: I am not exactly sure what the first sentence in the caption means. Do you mean step heights away from the beginning of the dune? This could be clarified perhaps.

    o *Yes, we mean step heights downstream of the upstream trough. We have added this for clarity.*

- 195: I suppose you want to say $q_s$ is the bedload transport rate (rather than sediment), since you mentioned Exner in the absence of suspended sediment exchange.

    o *We have changed the definition of $q_s$ to be bedload transport rather than sediment transport.*

- 203: Maybe this conundrum is a bit overstated. Nonlinearities in the relationship between the bed slope and the sediment transport rate produce dynamic bedforms with statistically stable characteristics. The linear stability has been analyzed (Patterns of Dirt, Neils Balmforth 2002), and initial instabilities have been attributed to noise in the sediment transport rate (Bohorquez et al near 2015)

  - *We have removed the three or so sentences that set up this conundrum and rather simply discuss the pattern of streamwise sediment transport decreasing at the crest of the bedform.*

- 214: " A linear increase in transport is necessary for bedforms to retain a two-dimensional geometry while translating downstream". Jerolmack and Mohrig (2005) maintain quasi two dimensional bedforms with a nonlinear Exner equation (see their Eq. 8 and Fig 4). How do you reconcile their successful nonlinear formulation of bedform dynamics with your statement about linearity ($q_s \sim x$) being required? It seems to me so long as there is no cross-stream variation in sediment transport trends (i.e. $q_s(x,y)=q_s(x)$), it does not matter how $q_s$ scales with $x$, provided it has a positive gradient on the stoss side of the bedform (i.e., degradational) and a negative gradient at the crest and on the lee side of the bedform (i.e., depositional)-- this will produce a downstream migrating bedform with some two dimensional profile (which depends on the specific gradient values), regardless of whether these gradients are constant or not. Maybe I am missing the point - in this case a clarification may benefit others. I otherwise agree that any lateral variation in $q_s$ can destabilize a 2D bedform.

  - *We have rephrased the statement to, "A consistent linear increase in transport rate through time is necessary, however, to maintain a spatially and temporally constant erosion rate." The point we are trying to make is that a linear increase in transport rates is the only scenario in which erosion on the stoss side is equal at all locations. Yes, two-dimensional bedforms can be formed from other, non-linear patterns of sediment transport but they do not yield a constant erosion rate along the entirety of the bedform.*

- 233: $x^a$ is algebraically, not exponentially - $a^x$ - this requires corrections in the figures and elsewhere in the text

  - *We have changed "exponentially" to "algebraically" in the text.*

- 292: typos

- 294: "Considering the results presented herein, we suggest two potential mechanisms that drive the transition from two-dimensional to three-dimensional bedform geometries: (1) splat events near flow reattachment and (2) localized, nonlinear increases in bedload transport rates along the stoss side of the bedform." Jerolmack and Mohrig (2005) show that stochasticity in sediment transport drives transitions to three dimensional geometries.

  - *We have added this reference to the statement.*

- "(splat events near flow reattachment) and (localized, nonlinear increases in bedload transport rates) may be genetically linked, and we suggest that (1) could drive (2)." Can you investigate the timescales separating sequential splat events and

compare these with the timescales separating what you believe are splat-associated transport events? How do they compare?

- o *We thank the reviewer for this thoughtful suggestion. Although it is possible to automate and estimate the timescales separating {1 4} events, it is rather difficult to estimate the timescales separating splat events. At this time, we would need to identify each individual splat event manually—a process that would be fraught with user bias/error. Additionally, we would need to identify enough splat events to provide temporal estimates. We are unsure if we would be able to identify an adequate timeseries of splat events to be statistically significant.*

- Isn't it strange that Reynolds stresses and mean bedload transport are anticorrelated? Can you clarify this? Are the Reynolds stresses near the bed irrelevant to the mean bedload transport rate over dunes?

  - o *There is high shear at the top of the separation bubble which gives rise to high Reynolds stresses. This strip of high shear and Reynolds stress is near the bed at the point of flow reattachment and further downstream. Venditti et al. (2007) found elevated Reynolds stresses near flow reattachment. This does not mean that the bed stress (the actual mean force per unit area on the bed grains) is the highest here because the main balance in the Navier-Stokes equations is between flow acceleration and the downstream pressure gradient. The near-bed flow velocity is increasing from the point of reattachment*

*We thank the reviewer for their thoughtful and thorough review of the manuscript. Below are our responses to major and minor comments in green italic.*

**Referee #2 Comments:**

This paper presents experimental results regarding the turbulent structure and bedload transport pattern over a two-dimensional ripple using LDV velocity measurement and camera-based particle tracking techniques. More specifically, the authors showed the presence of a "splat event" at the flow separation and reattachment point behind ripple crest, and discussed its importance for the bedload transport and morphodynamics of ripple. Understanding the relationship between the flow turbulence and bedload transport provides some important insight into the mechanics and dynamics of bedforms; however, this has not been deeply understood because of the difficulty of measurement. Therefore, this paper will be a nice contribution to our understanding of fluvial morphodynamics, and the paper's topic well fits the scope of ESurf. I would like to point out some unclear points about the experimental method and analysis and some interpretations of the results as follows.

Detailed comments:

1. The discussion on the bedload transport pattern and bedform geometry needs to be checked carefully. For example, sentence like Lines 134-135 sounds OK for a necessary condition for downstream migration of bedforms, but is not really applicable to explain two-dimensional features. If there is no variation of streamwise bedload transport rate in cross-stream direction, any streamwise distribution of bedload transport (either linear or nonlinear) could sustain two-dimensional bedform. So, I am not sure the discussion at Lines 211-238 makes sense. To discuss above, the authors presented Exner equation (2). Normally, \partial q_s/\partial y is not included in the Exner equation. What is the control volume for deriving this equation? Please check carefully since referee #1 also pointed out a similar issue regarding Equation (2).

   - *We have modified section 4.1 to reflect the comments noted here and by Reviewer #1. We no longer assert that a linear increase in transport over the bedform is necessary to sustain 2D bedforms. We have also modified our Exner equation to no longer include \partial q_s/\partial y as we do not think it benefits the discussion or the points we are trying to make.*

2. The importance of splat event on the bedload transport over ripple bed should be clearly explained. This is because that one of the main contributions of this paper may be that the authors use the concept of splat event, which was previously pointed out by the same team (Leary and Schmeeckle, 2017, JGR) using a flume with backward facing step, to understand real bedform fields. In addition, I am not sure about the discussion on the importance of the splat event to the three-dimensional bedform mentioned at Lines 294-297. The authors suggested an interesting mechanism, but I am unsure whether or not the experimental result supports this. The bedload transport feature at downstream of the crest indeed shows complex cross-stream pattern, but even though bedload transport pattern at more downstream shows a strong streamwise dominated feature. This may be because of the condition selected in this study: under the same but purely movable

bed condition, the bedform presented by Nelson et al. (2011) seems to show two-dimensional feature. More clear discussion with some literature review will be beneficial to highlight the discussion here.

- *We have changed the title of this section (section 4.3) to be "The Potential Role of Splat Events to Bedform Three Dimensionality" as we think it is a better representation of the discussion within that section. Section 4.3 is meant as a discussion of future hypotheses to be explored. We acknowledge that this present study does not explicitly support these hypotheses with results, hence why we discuss these concepts in the discussion section.*

3. There are some unclear points in the experimental work and its analysis. For example, the LDV measurement are performed along the centerline of the flume above 1 and 3 mm from the bed. How did the authors calculate mean velocity and Reynolds stress using this measurement? This is unclear since one velocity quantity is only shown at single streamwise location. Is there any difference of velocity feature at 1 and 3 mm above the bed? Also, Line 127 mentioned cross-stream fluid velocity, but I think there is no LDV measurement in cross-stream velocity. Please confirm. In addition to the flow measurement, a detailed explanation of bedload transport measurement should be added in the text. As referee #1 pointed out, the sampling time is important for calculating bedload flux. It might be better to explain the direction, length and velocity of sediment particles at the method section not around Line 251 since these are also important quantities to characterize the bedload transport pattern.

*Mean velocity data and Reynolds stress were calculate using the 3mm above the bed LDV data only. We collected data at 1mm from the bed but did not use it for any calculations. We have specified this in the manuscript.*

*We have fixed the typo at line 127. We meant to write streamwise and vertical fluid velocities. We thank the reviewer for catching our mistake.*

*We have added the sampling time used for calculating bedload transport rates to section 2.2, which describes how bedload transport rates were calculated. We have updated this section to provide more information about how bedload transport was measured.*

**Line-by-line comments:**

Lines 77 and 80: 3 and 2 should be superscript.

*We have fixed these typos.*

Line 81: This flow depth is an averaged value over 1 ripple or others? And how did the authors control the water depth. Please confirm.

*Flow depth was averaged over the single test ripple. We controlled the water depth with a weir at the downstream end of the flume.*

Table 1: The grain shape is assumed as a spherical to conovert the unit from grains/(cm*s) to cm^2/s? Also, did the authors check this value reasonably like by comparing sediment supply rate or sediment transport rate at the downstream end.

*Yes, the grain shape was assumed to be spherical. Since only the test ripple was loaded with sand, there was no supply rate or downstream sediment transport rate to compare it to.*

Line 122: "being downstream at the crest" will be better.

We are happy with how we have phrased this. We reference everything in step height downstream of the trough and prefer to not mix the two.

Lines 259-260: Is there any reference for previous study?

*We have added a reference to Leary & Schmeeckle 2017.*

Line 269: What is the definition of concentration of sediment?

*Radice (2021) uses the term "concentration" to refer to a relative mass of moving sediment. Specifically, C = (N x W)/(A x d) where N is the number of particles moving over a measuring area, A, over the time interval that separates two frames. W is the volume of one particle.*

Line 303: in- creased should be increased

*We have fixed this typo.*

---

## Author Response (AR2)

*We thank the Associate Editor for their thoughtful comments. Please find our responses to those comments below in green italic.*

Thanks for submitting your revised version of this paper and for addressing the reviewer's comments. I am happy to recommend that the paper is accepted for publication subject to some minor revisions as outlined below. Line numbers refer to lines in the tracked changes version of the paper.

36: Explain what a quadrant 4 event is? You define them later on, but an earlier explanation might be useful for some readers.
*We have added a brief explanation of quadrant 4 and 1 events to the introduction.*

Throughout the intro, check reference formatting. A number have missing brackets around the dates.
*We have fixed these errors.*

47/49/61: Address question marks in the text.
*These question marks only appear in the track changes file. They represent references not found in the version of the .bib file we were using. We have fixed this mistake in the track changes file as well as the newest version of the manuscript.*

50 and 56: Make it clear throughout this section if the work you are referring to was physical or numerical experiments.
*We have added the terms "physical" and "numerical" where it is unclear.*

67: Do you mean the difference between transport over a step and dune, or between lee and stoss sides?
*We mean the difference between transport over a step versus and dune. We have added more text to make this distinction more obvious.*

Fig 1: Text in the insert is a bit small.
*We have revised the figure and made the text in the inset larger.*

93: It wasn't entirely clear to me from the text if the sand was added just to the stoss side, or to the stoss and lee sides.
*Just the stoss side was loaded with sediment. We have added this clarification to the text.*

95: 3 should be in superscript not subscript. Throughout this section, make sure that units are not in italic font and that there is a space between numbers and units.
*We have fixed this typo as well as revised all units to be unitalicized and have a space between numbers and units.*

141: Add a brief explanation of the two different methods – it wasn't clear to me why they gave opposite results in a later figure.

*We have added a brief explanation of the two different methods of calculating flow exuberance.*

Fig 4: Add a dashed line to show the flow reattachment location as in the previous figure.

*We have added a dashed line to indicate flow reattachment.*

191: Do you mean quadrant 1 and 3 events?

*No, we mean quadrant 2 and 4 events as those are associated with low/near-zero values of exuberance.*

Fig 10: Axes labels are very small.

*We have revised the figure to make the axis labels bigger.*

343: The results up to this point highlighted the occurrence of quadrant 2 & 4 events. What is the evidence for quadrant 1 events occurring?

*The evidence for quadrant 1 events comes largely from previous research, namely Leary & Schmeeckle (2017), which we cite at this location. Essentially, from analysis of numerical experiments of fluid and sediment dynamics downstream of a backstep, the authors found that the octants associated with splat events were overwhelmingly octants 1, 4, and -4. We have added a sentence to the conclusions that addresses Quadrant 2 events: "Analysis reported herein also suggests the importance of quadrant 2 events near flow reattachment but it is unclear how they play a role in splat events."*